# Evaluation and Heritability Analysis of the Seed Vigor of Soybean Strains Tested in the Huanghuaihai Regional Test of China

**DOI:** 10.3390/plants12061347

**Published:** 2023-03-16

**Authors:** Hao Cheng, Mengyuan Ye, Tingting Wu, Hao Ma

**Affiliations:** 1National Key Laboratory of Crop Genetics & Germplasm Enhancement and Utilization, Nanjing Agricultural University, Nanjing 210095, China; 2018201051@njau.edu.cn (H.C.);; 2Institute of Crop Science, Chinese Academy of Agricultural Sciences, Beijing 100081, China

**Keywords:** soybean, seed vigor, genotype, environment, interaction, generalized heritability, regional test

## Abstract

High seed vigor is a prerequisite for high and stable yield. At present, seed vigor is not considered a goal of soybean breeding in China. Therefore, the seed vigor status of soybean strains is unclear. In this study, the seed vigor of 131 soybean strains included in the Huanghuaihai regional test in 2019 was measured using the artificial accelerated aging method. Significant (*p* < 0.01) differences in seed vigor were found, with a coefficient of variation of four vigor indexes being more than 48%. The strains with high vigor only accounted for 28.24%, indicating the seed quality of the tested strains was generally not high. Furthermore, the seed vigor of four representative strains selected from each of three vigor types was evaluated using multiyear and multilocation experiments. The seed vigor indexes of high-vigor-type strains were the most stable, while medium- and low-vigor-type strains varied under different environmental conditions. The generalized heritability of the four vigor indexes of the three vigor types was more than 51% and in the order of high vigor type > low vigor type > medium vigor type. Our results suggested that the genotypes of high-vigor-type strains have a greater influence on seed vigor, so to ensure varieties with high seed vigor, this criterion should be prioritized in soybean breeding programs in China.

## 1. Introduction

Seed vigor refers to the comprehensive potential to determine rapid and orderly seed emergence and normal seedling growth under a wide range of field conditions and is an important physiological index to measure seed quality [1]. High-vigor seeds germinate and emerge uniformly and rapidly under an extensive range of field conditions and display resilience to adverse environmental conditions. In contrast, seeds with low vigor can germinate under favorable conditions but germinate slowly, and their emergence is irregular or even limited under adverse conditions [2]. Therefore, seed vigor can be further described as fully mature, full, healthy, disease-free, intact, and non-dormant seeds with good storage tolerance. High-vigor seeds implement a stress avoidance strategy under deteriorating soil conditions through rapid seedling establishment while soil conditions are adequate [3,4].

Seed vigor is determined by three main factors: genetic factors, environmental conditions during seed development, and storage conditions. Genetic factors determine the seed vigor intensity, and environmental conditions determine the seed vigor expression degree [5]. Many studies have shown that the difference in the seed vigor level is objective and is largely determined by the genotype. Large differences in seed vigor have been found among different crop species or different strains of the same species, and seed vigor can be inherited [6,7,8]. Environmental conditions (including temperature, light, water, mineral nutrition, and pests and diseases) affect seed vigor throughout the crop [9,10]. In the actual production process, high-vigor seeds are an important guarantee for crops to achieve high and stable yield. High-vigor seeds have less seed use, a higher emergence rate, better resistance to adverse events, and better growth in the field [11,12,13]. Although low-vigor seeds can germinate under suitable conditions, they have a large amount of seed use, late emergence, a low emergence rate, and poor stress tolerance [14].

Soybean is one of the most important economic and food crops in the world. Soybean strains used in production are usually conventional strains. For a long time, the breeding objectives for soybean have mainly focused on high yield and quality [15,16], while high vigor is not considered one of the breeding objectives, especially in China. Therefore, it is often unclear whether the bred soybean strains have high seed vigor. In this study, to examine whether there are differences in seed vigor among local adapted strains, 131 soybean strains that were included in the regional test in Huanghuaihai, China, were evaluated and classified using the artificial accelerated aging method [1]. Further, representative strains selected from three vigor levels, namely high, medium, and low, were used to reveal the changes in seed vigor and clarify the influence of genetic and environmental factors on seed vigor using multiyear and multilocation experiments. Our results could provide a theoretical basis and reference for the formulation of soybean breeding objectives and selection criteria for future breeding programs in China.

## 2. Results

### 2.1. Evaluation and Cluster Analysis of the Seed Vigor of Soybean Strains Tested in the Regional Test

#### 2.1.1. Evaluation and Cluster Analysis of the Seed Vigor of 131 Soybean Strains

The seed vigor of 131 soybean strains that were included in the regional test in Huanghuaihai, China, in 2019 was investigated using the artificial accelerated aging method, and the results are listed in Table 1. The averaged germination potential (GP), germination rate (GR), germination index (GI), and vigor index (VI) of the 131 soybean strains were 49.26%, 57.49%, 7.87, and 1.63, respectively. The coefficient of variation of the VI was the largest, and the coefficient of variation of the GR was the smallest in the 131 soybean strains, with the coefficient of variation of all four vigor indexes being more than 48%. Moreover, there existed significant (*p* < 0.01) differences in the four vigor indexes among different soybean strains, which implied that there were significant difference in seed vigor among the strains included in the soybean regional test in Huanghuaihai, China, in 2019.

Furthermore, the 131 soybean strains were classified using the systematic clustering method, based on their vigor indexes. When the Euclidean distance was 5, all of them could be divided into three categories, and the mean values of the vigor indexes of each category were 2.75, 1.59, and 0.46, respectively. There existed significant (*p* < 0.05) differences in vigor indexes among the different categories. Among them, 37 strains fell into the high vigor level, accounting for 28.24% of all the 131 soybean strains; 61 strains belonged to the medium vigor level, accounting for 46.57%; and 33 strains fell into the low vigor level, accounting for 25.19% (Table 2, Appendix A). Notably, the strains with medium and low vigor levels accounted for 71.76% of all the 131 soybean strains.

#### 2.1.2. Evaluation and Cluster Analysis of the Seed Vigor of 34 Soybean Strains

To verify the results, 34 soybean strains included in the regional test in Huanghuaihai, China, in 2021 were used to investigate their seed vigor, and the results are listed in Table 3. The averaged GP, GR, GI, and VI of the 34 soybean strains were 68.32%, 76.56%, 10.64, and 2.36, respectively. The results of ANOVA showed that there were significant (*p* < 0.01) differences in the four vigor indexes among the 34 soybean strains, implying that there were also significant difference in seed vigor among the soybean strains included in the regional test in Huanghuaihai, China, in 2021.

Based on the vigor indexes, when the Euclidean distance was 5, the 34 soybean strains could also be divided into three categories, and the mean values of the vigor indexes of each category were 2.91, 1.93, and 0.97, respectively. There existed significant (*p* < 0.05) differences in vigor indexes among the different categories. Among them, there were 18 strains falling into the high vigor level, accounting for 52.94% of all the 34 soybean strains; 13 strains were at the medium vigor level, accounting for 38.24%; and 3 strains belonged to the low vigor level, accounting for 8.82% (Table 4, Appendix A). Our results indicated that there were still 47.06% soybean strains included in the regional test in Huanghuaihai, China, in 2021 that fell into the medium and low vigor levels.

Taken together, our results indicated that there existed significant (*p* < 0.01) differences in the seed vigor among soybean strains included in the regional test in Huanghuaihai, China, implying that the seed quality of the tested soybean strains was generally not high.

### 2.2. Analysis of the Seed Vigor of Soybean Strains Using Multiyear and Multilocation Experiments

#### 2.2.1. Change in Seed Vigor

From each of the high, medium, and low vigor levels of the 131 soybean strains, 4 representative strains were randomly selected; a total of 12 soybean strains were used for multiyear and multilocation experiments, and their vigor indexes are listed in Table 5.

The seed vigor of 12 representative soybean strains with different vigor levels harvested from the multiyear and multilocation experiments was investigated, and the results are listed in Table 6. For the 12 soybean strains, the averaged GP, GR, GI, and VI were 66.75%, 78.03%, 11.84, and 2.10, respectively, and the coefficient of variation of the four vigor indexes was 30.04%, 20.41%, 21.47%, and 24.88%, respectively. For the different vigor levels, the average values of the four vigor indexes were in the order of high vigor type > medium vigor type > low vigor type. The coefficients of variation of the GI were in the order of low vigor type > medium vigor type > high vigor type, while the coefficients of variation of the GR, GP, and VI were in the order of medium vigor type > low vigor type > high vigor type. The coefficients of variation of the four vigor indexes of the high-vigor-type strains were the lowest. Our results indicated that the seed vigor indexes of high-vigor-type strains were the most stable in the multiyear and multilocation experiments, while the medium- and low-vigor-type strains were easily affected by environmental conditions and their seed vigor indexes were unstable.

#### 2.2.2. Effects of Genetic Factors, Environmental Factors, and Interaction between Genetics and the Environment on Seed Vigor

The vigor indexes of 12 soybean strains with different vigor types in the multiyear and multilocation experiments were analyzed with analysis of variance (ANOVA), and the effects of genetic factors, environmental factors, and their interaction on seed vigor were calculated (Table 7). The results showed that the genotypes, environments, and their interactions had significant (*p* < 0.01) effects on the four seed vigor indexes of all the three vigor types, but there were still differences in the degree of influence. For high-vigor-type strains, the explained variation ratios (%) of the genotypes of all the four vigor indexes were higher than those of the environments and their interactions, indicating that the four vigor indexes of the high-vigor-type strains were controlled by genotypes and not vulnerable to environmental influence. For the medium-vigor-type strains, the explained variation ratios (%) of the genotypes of the GP and VI were higher than those of the environments and their interactions, while the explained variation ratio (%) of the environments of the GR was higher than the ratios of the genotypes and their interactions, and more interestingly, the explained variation ratio (%) of the interactions of the GI was higher than the ratios of the genotypes and the environments. For example, the seeds of SY01 harvested in Shandong in 2020 were of medium vigor, while those harvested in other years and regions were of high vigor. The seeds of SY31 harvested in 2020 in Anhui, 2021 in Anhui, and 2021 in Shandong were of medium vigor, while those harvested in 2020 in Shandong were of low vigor. The seeds of SY57 harvested in Anhui in 2020 and in Shandong in 2021 were of medium vigor, whereas those harvested in Shandong in 2020 and in Anhui in 2021 were of high vigor. The seeds of SY58 harvested in Anhui in 2020 and in Shandong in 2020 and 2021 were all of medium vigor, whereas those harvested in Anhui in 2021 were of high vigor. For the low-vigor-type strains, the explained variation ratios (%) of the genotypes of the GR and VI were higher than those of the environments and their interactions, while the explained variation ratio (%) of the interactions of the GI was higher than the ratios of the genotypes and the environments, and the effects of the genotypes and the environments on the GP were almost the same. All these results indicated that the four vigor indexes of the medium- and low-vigor-type strains showed partial instability due to environmental influence. However, overall, the influence of the genotypes on the four vigor indexes of the 12 soybean strains was greater than those of the environments and their interactions.

Taken together, our results indicated that soybean seed vigor is mainly controlled by the genotype, and compared to high-vigor-type strains, the seed vigor of medium- and low-vigor-type strains is more easily affected by environmental factors.

#### 2.2.3. Heritability of Seed Vigor

The generalized heritability of the seed vigor of strains of different vigor types was calculated, and the results are shown in the Table 8. The generalized heritability of the four vigor indexes of the three vigor types was more than 51%, further indicating that soybean seed vigor is controlled by heredity. The order of the generalized heritability of the four vigor indexes was high vigor type > low vigor type > medium vigor type.

## 3. Discussion

Seed vigor is one of the important indicators for evaluating seed quality. Seeds with high vigor have obvious growth advantages and production potential [17]. Studies have shown that the use of high-vigor seeds can increase crop yield by more than 35% [18,19]. The differences in seed vigor are widespread across different strains of the same crop [20,21], and seed vigor can be divided into three vigor levels: high, medium, and low [22]. Hao measured the seed vigor of 419 soybean germplasm resources and found that only 34.8% of soybean strains had a GR higher than 85% [23]. In this study, significant (*p* < 0.01) differences in seed vigor were found among the 131 soybean strains tested in the Huanghuaihai regional test, China, in 2019, with the variation coefficient of the four vigor indexes being more than 48%. The medium- and low-vigor-type strains accounted for 71.76% of the 131 strains, while the strains belonging to the high vigor type only accounted for 28.24%. The highest GR and VI of the strains of the high vigor type reached 100% and 3.38, respectively, while the lowest GR and VI of the strains of the low vigor type were lower, at 4% and 0.09, respectively. Moreover, there were still 47.06% soybean strains included in the regional test in Huanghuaihai, China, in 2021 that fell into the medium and low vigor levels. All the results indicated that the seed quality of the soybean strains tested in the regional test in Huanghuaihai, China, is generally not high. This is mainly because in the process of soybean breeding selection, breeders mainly focus on high yield, high quality, high resistance, and adaptation to mechanized production, while ignoring the selection of seed vigor, resulting in all the selected soybean strains not being of high vigor type. Strains with low vigor are likely to increase seed usage and cost and under poor environmental conditions even lead to reduced production. Strains with medium vigor, under suitable growth conditions, can produce high-vigor seeds. To ensure that the soybean breeding promotes high seed vigor, we suggest that this trait be prioritized.

The main influencing factors of seed vigor are genetic factors, external environmental factors during seed development, and the storage environment after harvest. Many studies have shown that both the genotype and the environment have a significant influence on seed vigor and the genotype is a determining factor in seed vigor [24,25], and high-vigor-type strains have a lower coefficient of variation of seed vigor under different environmental conditions [26]. Ma et al. indicated that the GP and GI of different wheat strains have high generalized heritability [27]. She et al. indicated that the generalized heritability of all the indicators of maize seed viability exceeds 50% [28]. In this study, the seed vigor of soybean strains of different seed vigor types was studied using multiyear and multisite experiments. The results showed that the four vigor indexes of the soybean strains of the high vigor type were higher than those of the strains of the medium- and low vigor types, and the coefficients of variation of the four vigor indexes of the high-vigor-type strains were lower than those of the strains of the medium and low vigor types. The seed vigor indexes of the high-vigor-type strains were the most stable in the multiyear and multilocation experiments, while the medium- and low-vigor-type strains were easily affected by the environments and their seed vigor indexes were unstable. Our results further indicated that the influence of genotypes on the four vigor indexes was greater than that of the environment and their interaction, implying that soybean seed vigor is mainly controlled by the genotype, and compared to high-vigor-type strains, the seed vigor of medium- and low-vigor-type strains is more easily affected by environmental factors. Moreover, the generalized heritability of the four vigor indexes of the soybean strains of the three vigor types was more than 51%, further indicating that soybean seed vigor is controlled by heredity.

Our results of the multiyear and multilocation experiments were consistent with a previous study, which demonstrated that high-vigor strains produce high-vigor seeds and vice versa [29]. In addition, the environment seems to play an important role in seed vigor in medium-vigor strains. This might be the reason that the order of the generalized heritability of the four vigor indexes was high vigor type > low vigor type > medium vigor type in multiyear and multilocation experiments. As the genotype x environment interaction had a significant effect on soybean seed vigor, especially on the seed vigor of medium-vigor-type strains, to ensure that the soybean strains promoted have high-seed-vigor characteristics, we suggest that the selection of high-seed-vigor traits should be listed as one of the important goals of soybean breeding and that the detection of seed vigor should be included in the detection items of soybean regional tests in China.

## 4. Materials and Methods

### 4.1. Materials

In total, 131 soybean strains that were included in the soybean regional test in Huanghuaihai, China, were collected and then planted in 2019 on the experimental farm of the Anhui Science and Technology University. Among them, 83 (numbered from SY01 to SY83 in the study) were provincial pre-test materials, 27 (SQ01–SQ27) were the materials of the provincial regional test, and 21 (GQ01–GQ21) were the materials of the national regional test. In addition, 34 soybean strains (BS01–BS34) that were included in the soybean regional test in Huanghuaihai, China, in 2021 were harvested at the regional test site at Shunyi, Beijing, China. There were no common strains between the regional tests in 2019 and 2021.

Further, 12 representative soybean strains with different vigor levels selected from the 131 strains were analyzed using multiyear and multilocation experiments on the experimental farm of the Anhui University of Science and Technology, Anhui Province, and at the Shengfeng seed industry base, Shandong Province, in 2020 and 2021.

All the harvested soybean seeds were stored at 4 °C for further testing.

### 4.2. Determination of Seed Vigor

Soybean seeds were randomly counted from the well-mixed seed samples after cleanliness analysis. For each treatment, three replicates were set and each replicate included 100 seeds. Soybean seeds were pre-soaked in 1% sodium hypochlorite solution for 15 min for surface disinfection and then washed three times with sterile water. At the end of the treatment, the seed samples were spread thinly and dried back to the original moisture content.

Seed vigor was determined using the artificial accelerated aging method (ISTA, 2015). Briefly, the well-prepared soybean seeds were placed in an aging chamber at 40 °C and 100% humidity for accelerated aging for 3 d. After aging, the treated seeds were subjected to the standard germination test. The germination potential (GP) was recorded on the 4th day and the germination rate (GR) on the 7th day, and the relevant germination index (GI) and vigor index (VI) were calculated according to the following formula, using the standard germination test as a control:$$\mathrm{GP}=\frac{\mathrm{number}\;\mathrm{of}\;\mathrm{germinated}\;\mathrm{seeds}\;\mathrm{on}\;4\mathrm{th}\;\mathrm{day}}{\mathrm{total}\;\mathrm{number}\;\mathrm{of}\;\mathrm{seeds}\;\mathrm{for}\;\mathrm{testing}}\times 100\%$$
$$\mathrm{GR}=\frac{\mathrm{number}\;\mathrm{of}\;\mathrm{germinated}\;\mathrm{seeds}\;\mathrm{on}\;7\mathrm{th}\;\mathrm{day}}{\mathrm{total}\;\mathrm{number}\;\mathrm{of}\;\mathrm{seeds}\;\mathrm{for}\;\mathrm{testing}}\times 100\%$$
$$\mathrm{GI}=\sum_{}^{}\left(\frac{\mathrm{Gt}}{\mathrm{Dt}}\right)$$
VI=GI×S
where Gt is the number of germinations per day, Dt is the number of days corresponding to Gt, and S is the dry weight of seedlings.

### 4.3. Data Analysis

Microsoft Excel 2010 was used to organize the data and calculate the mean values of each treatment trait, and SPSS 25.0 software was used for ANOVA, correlation analysis, cluster analysis, and graphing.

## 5. Conclusions

There were significant (*p* < 0.01) differences in seed vigor among the soybean strains included in the Huanghuaihai regional test in 2019, indicated that the seed quality of the tested strains was generally not high. The genotype, environment, and their interaction had significant effects on soybean seed vigor. Genotypes of the high-vigor-type strains had a greater influence on seed vigor than those of the medium- and low-vigor-type strains. The genetic differences were the main reasons for the differences in seed vigor among different soybean strains. Soybean seed vigor was mainly controlled by the genotype. The generalized heritability of seed vigor of the high-vigor-type strains was higher than that of the medium- and low-vigor-type strains. The soybean strains of the high vigor type produced high-vigor seeds, whereas the low-vigor-type strains produced low-vigor seeds, and the seed vigor of the medium-vigor-type strains significantly varied with the environment. To ensure that the soybean varieties promoted have high-seed-vigor characteristics, the selection of seed vigor traits should be listed as an important goal of soybean breeding and the detection of seed vigor should be included in the detection items of soybean regional tests in China.

## Figures and Tables

**Table 1 plants-12-01347-t001:** Variation in seed vigor indexes of 131 soybean strains included in the regional test in Huanghuaihai, China, in 2019.

Statistical Items	GP (%)	GR (%)	GI	VI
Mean	49.26 **	57.49 **	7.87 **	1.63 **
Standard error	27.40	27.97	4.33	0.94
Maximum	96.00	100.00	15.99	3.38
Minimum	1.00	4.00	0.35	0.09
Range	95.00	96.00	15.64	3.29
Variation coefficient (%)	55.62	48.65	54.95	57.95

Note: ** represented a significant level of difference at the 0.01 level.

**Table 2 plants-12-01347-t002:** Number of genotype clusters and their mean values of the vigor indexes of 131 soybean strains included in the regional test in Huanghuaihai, China, in 2019.

Statistical Items	High Vigor Level	Medium Vigor Level	Low Vigor Level
Number of clusters	37	61	33
GP (%)	84.38	48.26	12.33
GR (%)	90.65	59.00	18.42
GI	13.11	7.73	2.33
VI	2.75 a	1.59 b	0.46 c

Note: Different lowercase letters in the same line indicate significant differences at the 0.05 level.

**Table 3 plants-12-01347-t003:** Variation in the seed vigor indexes of 34 soybean strains included in the regional test in Huanghuaihai, China, in 2021.

Statistical Items	GP (%)	GR (%)	GI	VI
Mean	68.32 **	76.56 **	10.64 **	2.36 **
Standard error	19.03	16.89	2.99	0.77
Maximum	86.00	94.00	15.85	3.48
Minimum	20.00	30.00	3.08	0.61
Range	66.00	64.00	12.78	2.87
Variation coefficient (%)	27.76	22.18	28.29	32.65

Note: ** represented a significant level of difference at the 0.01 level.

**Table 4 plants-12-01347-t004:** Number of genotype clusters and their mean values of the vigor indexes of 34 soybean strains included in the regional test in Huanghuaihai, China, in 2021.

Statistical Items	High Vigor Level	Medium Vigor Level	Low Vigor Level
Number of clusters	18	13	3
GP (%)	82.06	60.08	20.67
GR (%)	88.50	69.38	31.33
GI	12.66	9.31	3.51
VI	2.91 a	1.93 b	0.97 c

Note: Different lowercase letters in the same line indicate significant differences at the 0.05 level.

**Table 5 plants-12-01347-t005:** Seed vigor indexes of 12 representative soybean strains with different vigor levels.

Vigor Level	Variety No.	GP (%)	GR (%)	GI	VI
Low	SY07	39.22	51.35	6.51	1.14
	SY12	34.95	39.17	6.68	0.93
	SY18	40.17	57.62	7.26	1.58
	SY19	31.35	47.38	6.82	1.36
	Mean	36.42	48.88	6.82	1.25
Medium	SY01	49.44	86.44	10.47	1.84
	SY31	70.58	84.58	13.02	1.71
	SY57	61.69	76.47	11.09	1.62
	SY58	56.97	64.28	9.83	1.85
	Mean	59.67	77.94	11.10	1.76
High	SY25	86.24	90.47	13.52	2.57
	SY34	96.37	97.55	15.99	3.22
	SY64	93.69	96.17	14.58	3.07
	SQ20	88.48	93.35	15.01	2.69
	Mean	91.20	94.39	14.77	2.89

**Table 6 plants-12-01347-t006:** Variation in the seed vigor of 12 representative soybean strains with different vigor levels in multiyear and multilocation experiments.

Vigor Level	Statistical Items	GP (%)	GR (%)	GI	VI
Low	Mean	44.11	61.28	9.03	1.59
	Standard error	6.80	9.55	1.49	0.26
	Maximum	56.00	75.00	10.82	1.89
	Minimum	31.00	39.00	6.51	0.93
	Range	25.00	36.00	4.31	0.96
	Variation coefficient (%)	15.42	15.59	16.46	15.55
Medium	Mean	67.85	78.36	12.03	2.07
	Standard error	12.77	12.22	1.29	0.35
	Maximum	87.00	95.00	14.38	2.83
	Minimum	41.00	46.00	9.83	1.62
	Range	67.85	78.36	12.03	2.07
	Variation coefficient (%)	18.82	16.59	10.75	16.90
High	Mean	88.18	93.89	14.41	2.63
	Standard error	5.36	2.44	1.04	0.33
	Maximum	96.37	97.55	16.15	3.22
	Minimum	80.00	90.00	12.21	2.14
	Range	16.37	7.55	3.94	1.08
	Variation coefficient (%)	6.07	2.60	7.20	12.62
Total	Mean	66.75	78.03	11.84	2.10
	Standard error	20.05	15.93	2.54	0.52
	Maximum	96.37	97.55	16.15	3.22
	Minimum	31.35	39.17	6.51	0.93
	Range	65.02	58.38	9.64	2.29
	Variation coefficient (%)	30.04	20.41	21.47	24.88

**Table 7 plants-12-01347-t007:** Effects of genetic and environmental factors on the seed vigor of soybean.

Vigor Type	Vigor Indexes	Source of Variation	Sum of Squares	Mean Square	Explained Variation Ratio (%)
High	GP (%)	G	613.26	68.14 **	46.10
		E	491.32	163.77 **	36.94
		G × E	225.56	75.19 **	16.96
	GR (%)	G	237.07	26.34 **	62.30
		E	55.87	18.62 **	14.69
		G × E	87.58	29.19 **	23.02
	GI	G	28.67	4.30 **	60.30
		E	14.20	4.73 **	25.81
		G × E	12.14	2.71 **	13.89
	VI	G	3.28	1.09 **	74.86
		E	0.68	0.23 **	14.45
		G × E	0.43	0.05 **	9.70
Medium	GP (%)	G	265.80	121.93 **	40.35
		E	259.57	19.86 **	28.20
		G × E	201.17	33.46 **	31.45
	GR (%)	G	3393.20	377.02 **	32.25
		E	4244.82	414.94 **	48.58
		G × E	1856.38	618.79 **	19.17
	GI	G	34.64	11.55 **	37.11
		E	15.03	5.01 **	16.11
		G × E	43.66	4.85 **	46.78
	VI	G	3.79	0.60 **	41.13
		E	2.48	1.16 **	21.05
		G×E	3.20	0.36 **	37.82
Low	GP (%)	G	694.80	77.20 **	43.58
		E	677.05	225.69 **	42.46
		G × E	222.54	74.18 **	13.96
	GR (%)	G	674.22	74.91 **	44.67
		E	583.62	194.54 **	38.67
		G × E	251.42	83.81 **	16.66
	GI	G	17.36	5.79 **	22.20
		E	17.54	5.84 **	22.40
		G × E	43.36	4.82 **	55.40
	VI	G	0.762	0.287 **	40.59
		E	0.545	0.148 **	37.00
		G × E	0.466	0.052 **	22.41
Total	GP (%)	G	487.92	443.57**	68.91
		E	205.19	135.06 **	20.74
		G × E	178.67	172.08 **	10.35
	GR (%)	G	235.80	203.55 **	58.23
		E	114.18	380.59 **	24.00
		G × E	107.88	153.90 **	17.77
	GI	G	297.60	27.06 **	63.31
		E	88.37	6.79 **	14.33
		G × E	152.07	4.61 **	22.35
	VI	G	13.09	1.19 **	51.54
		E	6.68	2.23 **	26.28
		G × E	5.64	0.17 **	22.18

Note: ** represented a significant level of difference at the 0.01 level. G, E, and G × E indicate genotypes, environments, and their interaction, respectively. Explained variation ratio (%) = SS_variation_ × 100/(SS_total_ − SS_error_ − SS_block_).

**Table 8 plants-12-01347-t008:** Generalized heritability analysis of the seed vigor of strains of different vigor types.

Statistical Item	Vigor Types	Generalized Heritability (h^2^)
GP (%)	High	78.73
	Medium	51.79
	Low	68.98
GR (%)	High	82.26
	Medium	57.40
	Low	62.45
GI	High	96.03
	Medium	73.24
	Low	84.59
VI	High	97.89
	Medium	75.38
	Low	90.95

Note: $h^{2}=\frac{{MS}_{V}-{MS}_{e}}{{MS}_{V}+\left(r-1\right)\times {MS}_{e}}\times 100\%$. ${MS}_{V}$, ${MS}_{e}$, and r indicate the variety average, mean-square error, and the number of repetitions, respectively.

## Data Availability

Not applicable.

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
