# Peer review of "Evaluation and Heritability Analysis of the Seed Vigor of Soybean Strains Tested in the Huanghuaihai Regional Test of China"

_plants, 2023, doi:10.3390/plants12061347_

Round 1

Reviewer 1 Report

Line 62 to 65, the sentence needs to be improved. Do the authors mean: To determine whether there were differences in seed vigor among locally adapted varieties, 131 strains from the Huanghuaihai regional test panel were selected for evaluation.

Line 62 to 71, the writing of this section, which states the objectives of the study, is not clear and needs to be retouched.  

Line 65, the “artificial accelerated aging method” was mentioned here. Background knowledge or references should be provided to explain what the method is.

Author Response

Dear Reviewer:

Thank you for your valuable and thoughtful comments and suggestions for our manuscript. We appreciate your excellent comments and suggestions. We have considered all of these suggestions and comments in revising our manuscript. The following lists our responses to your comments.

Best Wishes,

Hao Ma

State Key Lab of Crop Genetics and Germplasm Enhancement,

Nanjing Agricultural University,

Nanjing, Jiangsu Province 210095, China

Comments and Suggestions for Authors

  1. Line 62 to 65, the sentence needs to be improved. Do the authors mean: To determine whether there were differences in seed vigor among locally adapted varieties, 131 strains from the Huanghuaihai regional test panel were selected for evaluation.

Author’s Response: Thanks for the comment. We have revised the sentence.

  1. Line 62 to 71, the writing of this section, which states the objectives of the study, is not clear and needs to be retouched.

Author’s Response: Thanks for the comment. We have retouched the sentences.

  1. Line 65, the “artificial accelerated aging method” was mentioned here. Background knowledge or references should be provided to explain what the method is.

Author’s Response: Thanks for the comment. Added reference [1] to "artificial accelerated aging method".

Reviewer 2 Report

Recommendations are shown in the attached file.

Author Response

Thanks for the comments. We have made changes in revised manuscript.

Reviewer 3 Report

The manuscript is important for scientific community.

 Title: the title of the article is suitable.

Keywords:  are relevant for the topic of the paper.

Abstract: the abstract respect the editor’s conditions of editing work and present in brief the main purpose of the paper and the main results, who were carried out on the 131 strains of soybean, in China.

Introduction: the introduction part is fulfilled with information about the seed vigor and the influence of the different factors, genetic and environmental conditions during seed development, and storage conditions, and is linked with some scientific papers from the literature, representative for the topic.

Material and method: The authors describe the experimental location, treatments applied (three replicates were set and each replicate included 100 seeds). Was determined the germination potential (GP) and germination rate (GR) on 7th day, and the relevant indexes of germination index (GI) and vigor index (VI). Seed vigor was determined according to the artificial accelerated aging method 273 (ISTA, 2015). Microsoft Excel 2010 was used to organize the data and calculate the mean values of each treatment trait, and SPSS 25.0 software was used for ANOVA, correlation analysis, cluster analysis, and graphing.

Result: The result contain data about the significant effects of genotype, environment and their interaction on soybean seed vigor. Was demonstrated that the coefficients of variation of GI were the low vigor type > the medium vigor type > the high vigor type, while the coefficients of variation of GR, GP and VI were the medium vigor type > the low vigor type > the high vigor type. The seed vigor indexes of the high vigor type strain were the most stable in the multi-year and multi-location experiments, while the medium- and the low-vigor type strains were easily affected by environmental conditions and their seed vigor indexes were instable. The results presented correspond with other experiments.

Conclusions: The conclusions are very brief and compared to the volume of exposed, emphasizing the fact that the selection of seed vigor trait should be listed as an important goal of soybean breeding.

References: There are 28 bibliographic titles, good chosen and representative for the topic of the work, 17 bibliographic titles are from the last 10 years.

Author Response

Thanks for the comments.